# Steering the Metal Precursor Location in Pd/Zeotype Catalysts and Its Implications for Catalysis

Luc C. J. Smulders [1], Johan H. van de Minkelis [1], Johannes D. Meeldijk [1], Min Tang [1], Anna Liutkova [2], Kang Cheng [1], S. Tegan Roberts [3], Glenn J. Sunley [3], Emiel J. M. Hensen [2], Petra E. de Jongh [1] and Krijn P. de Jong [1],*

[1]  Materials Chemistry and Catalysis, Debye Institute for Nanomaterials Science, Utrecht University, Universiteitsweg 99, 3584 CG Utrecht, The Netherlands
[2]  Laboratory for Inorganic Materials and Catalysis, Department of Chemical Engineering and Chemistry, Eindhoven University of Technology, P.O. Box 513, 5600 MB Eindhoven, The Netherlands
[3]  Applied Sciences, bp Innovation & Engineering, BP plc, Saltend, Hull HU12 8DS, UK
*  Correspondence: k.p.dejong@uu.nl

**Abstract:** Bifunctional catalysts containing a dehydrogenation–hydrogenation function and an acidic function are widely applied for the hydroconversion of hydrocarbon feedstocks obtained from both fossil and renewable resources. It is well known that the distance between the two functionalities is important for the performance of the catalyst. In this study, we show that the heat treatment of the catalyst precursor can be used to steer the location of the Pd precursor with respect to the acid sites in SAPO-11 and ZSM-22 zeotype materials when ions are exchanged with $Pd(NH_3)_4(NO_3)_2$. Two sets of catalysts were prepared based on composite materials of alumina with either SAPO-11 or ZSM-22. Pd was placed on/in the zeotype, followed by a calcination-reduction (CR) or direct reduction (DR) treatment. Furthermore, catalysts with Pd on the alumina binder were prepared. CR results in having more Pd nanoparticles inside the zeotype crystals, whereas DR yields more particles on the outer surface of the zeotype crystals as is confirmed using HAADF-STEM and XPS measurements. The catalytic performance in both *n*-heptane and *n*-hexadecane hydroconversion of the catalysts shows that having the Pd nanoparticles on the alumina binder is most beneficial for maximizing the isomer yields. Pd-on-zeotype catalysts prepared using the DR approach show intermediate performances, outperforming their Pd-in-zeotype counterparts that were prepared with the CR approach.

**Keywords:** bifunctional catalysis; hydroconversion; palladium; zeolites; zeotypes; SAPO-11; ZSM-22

## 1. Introduction

Bifunctional catalysis is one of the many fields in which Avelino Corma has made significant contributions [1–3]. Bifunctional catalysts play a key role in petroleum refining for the production of fuels and chemicals from crude oil [4–6]. Furthermore, they are applied in the conversion of feedstocks such as Fischer–Tropsch wax [7,8], algae-based hydrocarbons [9,10] and plastic waste [11] into fuels and lubricants. Bifunctional hydroconversion catalysts consist of a metal (sulfide) function for (de)hydrogenation and an acid function for cracking or isomerization [12–15]. Parameters such as the type of acid function, the acidity of the acid materials, the type of metal, the metal-to-acid-site ($n_m/n_a$) ratio, proximity between the two types of sites and the porosity of the catalyst have an impact on the catalytic performance [14,16,17]. It is well known that the metal particles and the zeolite acid sites need to be in close proximity in order to obtain high catalytic activity [18,19].

Several studies on the proximity between metal and acid sites have been published, mainly evaluating the difference between placing the metal on alumina binder or on/in the zeolite crystals [16,20–23]. When the metal nanoparticles are located inside the zeolite crystals, their accessibility for larger or bulkier hydrocarbon molecules is limited, since these molecules have a low diffusivity inside the micropores [24,25]. This causes more

sequential reactions and hence (over)cracking of the feedstock. On the other hand, locating the particles slightly further away from the acid sites, i.e., on an alumina binder material or on the outer surface of the zeolite crystallites, results in improved selectivity towards the isomerized products [20,21,26]. Apart from zeolites, other zeotype materials can be used in bifunctional hydroconversion catalysts. For example, crystalline aluminium phosphates, SAPO-11 in particular, have been studied for this purpose [27–31].

Typically, salts containing cations such as $[Pt(NH_3)_4]^{2+}$ and $[Pd(NH_3)_4]^{2+}$ are added to a dispersion of a zeolite/zeotype in water in order to exchange protons with the metal precursor, after which a heat treatment is required to arrive at a catalyst with metal nanoparticles dispersed inside the zeolite crystallites [32–34]. From earlier work, we know that Pt dispersion in Pt/Y catalysts is influenced by the heat treatment conditions. A low calcination ramp (0.2 $^\circ$C/min) is required to prevent the formation of a mobile $Pt^0(NH_3)_2H_2$ complex and hence the formation of larger nanoparticles [34]. To our knowledge, the results obtained with these Pt/Y catalysts have never been correlated to metal nanoparticle location effects and the effect on catalysis has not been studied.

For applications such as hydroconversion, locating the metal nanoparticles (NPs) outside the zeolite micropores is beneficial. Other applications, however, such as propane dehydroaromatization [35] require the confinement of metal NPs inside the zeolite crystals. Furthermore, encapsulation of NPs inside zeolite materials can be used as a strategy to prevent catalyst poisoning [36]. Therefore, control over the metal nanoparticle location is important.

The location of metal nanoparticles can be studied using transmission electron microscopy (TEM) [20,37]. However, the disadvantages of TEM are the relatively small sample size, and that it only provides qualitative information about the metal nanoparticle location [26]. Therefore, a complementary and more quantitative characterization method is desirable. X-ray photoelectron spectroscopy (XPS) has proven to be relatively successful. It is a surface-sensitive technique, resulting in the fact that metal atoms located deep inside the zeolite hardly contribute to the signal, whereas metal atoms that are on the outer surface can be detected. Previously, this has been applied to the study of the location of Pd nanoparticles in carbon nanotubes [38] and silicalite-1 [39]. Oenema et al. [26] studied Pt/Al ratios in Pt/Y catalysts and used these data to provide quantitative information about the location of the Pt particles. One of the challenges they faced was the overlap between the Pt and Al signal, which is not problematic with Pd since the signal does not overlap with the other elements (Si, Al, P) that are to be determined using XPS.

In this work, we show that the heat treatment of the catalyst precursor can influence the nanoparticle location. Two sets of catalysts were prepared, one based on SAPO-11 and the other based on ZSM-22. These zeotypes both have 1D pore systems consisting of 10-membered rings and are typically used in hydroconversion research [21,22,29,31]. Each set of catalysts consisted of alumina/zeotype composite materials forming the basis of the Pd-on-alumina, Pd-zeotype-direct reduction (DR) and Pd-zeotype-calcination-reduction (CR) catalysts. These catalysts were denoted Pd-$\gamma$-Al$_2$O$_3$/SAPO-11, Pd-SAPO-11/$\gamma$-Al$_2$O$_3$ DR, Pd-SAPO-11/$\gamma$-Al$_2$O$_3$ CR, Pd-$\gamma$-Al$_2$O$_3$/ZSM-22, Pd-ZSM-22/$\gamma$-Al$_2$O$_3$ DR and Pd-ZSM-22/$\gamma$-Al$_2$O$_3$ CR. The Pd nanoparticle location was then studied using transmission electron microscopy (TEM) and X-ray photoelectron spectroscopy (XPS). The hydroconversion performance of the catalysts was assessed using *n*-heptane and *n*-hexadecane as the model feedstocks. This showed that the location of the metal nanoparticle clearly affected the performance in both sets of catalysts.

## 2. Materials and Methods

### 2.1. Raw Materials

SAPO-11 (H$^+$-form, Si/(Si+Al+P) = 0.15 at/at) and ZSM-22 (H$^+$-form, Si/Al = 32.5-40 at/at) were purchased from ACS Material. Boehmite was obtained from Harshaw. Tetraaminepalladium (II) nitrate (Pd(NH$_3$)$_4$(NO$_3$)$_2$) solution (10 wt% in H$_2$O) and ammonium hexachloropalladate ((NH$_4$)$_2$PdCl$_6$) were purchased from Sigma-Aldrich. *n*-Heptane (99+%, pure) and *n*-hexadecane (99%, pure) were purchased from ACROS Organics. Acetic acid

(glacial, 99+%) was purchased from Alfa Aesar. Hydrochloric acid (37 %) was purchased from Emsure. Silicon carbide (SiC, SIKA ABR I F70, grain size: 220 μm) was supplied by Fiven. $H_2$ 6.0, He 5.0 and $N_2$ 5.0 were obtained from Linde gas.

### 2.2. Catalyst Synthesis

### 2.2.1. Preparation of Catalysts with Pd on/in Zeotype Crystals

A schematic overview of catalyst synthesis is included in Figure S1 in the Supplementary Materials. SAPO-11 and ZSM-22 were calcined at 550 °C in a flow of synthetic air ($N_2/O_2$, 80/20, vol/vol) for 3 h. For the preparation of Pd-SAPO-11/$\gamma$-$Al_2O_3$ (DR) and Pd-ZSM-22/$\gamma$-$Al_2O_3$ (DR), 1 g of calcined SAPO-11 or ZSM-22 was dispersed in 300 mL Milli-Q water by stirring at 500 rpm for 1 h. The pH of the dispersion was 4.9 and 6.7 for SAPO-11 and ZSM-22, respectively. A solution with the desired concentration of $Pd(NH_3)_4(NO_3)_2$ was prepared by taking the required volume of the 10 wt% solution and diluting it in 50 mL Milli-Q water. This solution was then added dropwise to the zeotype dispersion, and the mixture was stirred for 3 h resulting in a pH of 5.9 and 6.6 for SAPO-11 and ZSM-22, respectively. Subsequently, the dispersion was filtered over vacuum and the filter cake was washed with 300 mL Milli-Q water. The filter cake was dried in an oven at 120 °C overnight. The pre-catalyst powders were then heat treated in a tubular oven. First, the powders were heated to 150 °C with a ramp of 2 °C/min to be dried in an $N_2$ flow of 80 mL/min for 1 h. To produce the DR catalyst, the powder was heated to 600 °C with a ramp of 2 °C/min in a hydrogen-rich atmosphere ($H_2/N_2$, 80/20, vol/vol) for 3 h. To prepare the regular Pd-in-zeotype catalyst, the powder was calcined ($N_2/O_2$, 80/20, vol/vol; 0.2 °C/min, 350 °C, 4 h) prior to reduction with the same reduction conditions. The Pd/zeotype (DR) powder was mixed with boehmite. During typical mixing, 1.428 g of boehmite was mixed with 1.2 mL of Milli-Q water and 0.042 mL of glacial acetic acid until complete wetting was reached using a mortar and pestle. This was followed with the addition of 1.00 g Pd/zeotype pre-catalyst and a few drops of Milli-Q water. It was mixed until a homogeneous paste was obtained. The paste was dried in an oven overnight at 120 °C after which it was calcined at 500 °C for 2 h (ramp 1 °C/min, $N_2/O_2$, 80/20, vol/vol).

### 2.2.2. Preparation of Pd-On-Alumina Catalysts

Pd-$\gamma$-$Al_2O_3$/SAPO-11 and Pd-$\gamma$-$Al_2O_3$/ZSM-22 were prepared using a strong electrostatic adsorption (EA) method as previously described in Cho and Regalbuto [40]. For this, 1 g of $\gamma$-$Al_2O_3$ (obtained from calcination of boehmite) was suspended in 300 mL Milli-Q water and stirred for 1 h using a mechanical stirrer. The suspension was acidified to a pH of 2.5 with 1M HCl solution. A precursor solution of 34.29 mg $(NH_4)_2PdCl_6$ in 33 mL Milli-Q water was added dropwise to the suspension and the suspension was stirred at 500 rpm for 3 h. The resulting suspension was then filtered over vacuum and the filter cake was washed with Milli-Q water. The filter cake was then dried in an oven at 120 °C overnight. This was followed by calcination ($N_2/O_2$, 80/20, vol/vol; 0.2 °C/min, 350 °C, 4 h) and then reduction ($H_2/N_2$, 80/20, vol/vol; 2 °C/min, 600 °C, 3 h) to obtain the Pd-$\gamma$-$Al_2O_3$ pre-catalyst. The Pd-$\gamma$-$Al_2O_3$ was then mixed with either SAPO-11 or ZSM-22. First, 1 g of calcined (550 °C, $N_2/O_2$, 80/20, vol/vol, 3 h) SAPO-11 or ZSM-22 was mixed with 1.2 mL Milli-Q water and 0.042 mL acetic acid using mortar and pestle, until complete wetting of the zeotype was reached. Then, 1 g of Pd-$\gamma$-$Al_2O_3$ was added and the paste was mixed with additional Milli-Q water until a homogeneous paste was obtained. This catalyst was then dried and calcined as also described for the Pd-zeotype/$\gamma$-$Al_2O_3$ catalysts.

### 2.3. Catalyst Characterization

### 2.3.1. Quantification of Metal and Acid Sites

Elemental analysis was performed at Mikroanalytisches Laboratorium Kolbe, Germany, with an inductively coupled plasma (ICP) optical emission spectrometer (PerkinElmer, Waltham, MA, USA) after sample dissolution according to their standard in-house procedures.

Temperature-programmed desorption of ammonia ($NH_3$-TPD) was performed to determine the number of acid sites. The measurements were performed on a Micromeritics AutoChem II equipped with a thermal conductivity detector (TCD) calibrated for ammonia. For a typical measurement, 100–110 mg of catalyst was dried in a He flow for 1 h at 550 °C with a ramp of 10 °C/min. The temperature was then lowered to 100 °C and ammonia was introduced (10 vol % in He) in a pulse-wise manner. After oversaturation was reached, the physisorbed ammonia was removed by flowing He for 2 h at 100 °C. This was followed by monitoring the desorption of ammonia up to 550 °C with a ramp of 10 °C/min.

Another technique that was applied to obtain information about the acid sites was Fourier Transform Infrared (FT-IR) Spectroscopy. FT-IR was performed in transmission mode on a Thermo iS5 instrument equipped with a DTGS detector. Approximately 15 mg of sample was pressed into a self-supported wafer, which was then placed in a sealed cell with calcium fluoride windows. The wafers were first dried under high vacuum at 550 °C (ramp 10 °C/min) for 2 h. After cooling down to a temperature of 30–40 °C, a spectrum was taken. For each spectrum, 16 scans were taken with a resolution of 4 $cm^{-1}$. This was followed by the introduction of pyridine (Sigma-Aldrich, 99.8%, probe molecule) at a pressure of 20 mbar, during which spectra were recorded every 2 min for 30 min starting 2 min after initial introduction of the pyridine. A high vacuum of $10^{-5}$ mbar was applied for 30 min in order to desorb the pyridine. This was followed with an increase in temperature to 150 °C (ramp 2 °C/min) and the temperature was kept at this point for 30 min, after which a spectrum was recorded. This was followed by heating the wafer to 250 °C with a ramp of 10 °C/min, after which an additional spectrum was taken that was used to assess the acidity of the samples. Quantification of acid sites, C (mmol $g^{-1}$), was performed using band integration of the peaks at 1545 $cm^{-1}$ (Brønsted acid sites, BAS) and 1453 $cm^{-1}$ (Lewis acid sites, LAS) in the spectrum after pyridine desorption at 250 °C utilizing Equation (1). The integral under the curve is represented by $A$ ($cm^{-1}$). The results were corrected for the specific molar absorption coefficients and the mass and radius of the wafers. Molar absorption coefficients ($A_0$) of 1.67 cm/μmol (BAS) and 2.22 cm/μmol (LAS) were used for quantification [41–43]. The mass of the wafer (mg) per $cm^2$ through which the IR beam was sent was represented by $\rho$. The stoichiometry of pyridine adsorption was assumed to be 1 molecule of pyridine molecule per LAS and 1 molecule of pyridine per BAS.

$$C = \frac{A}{A_0 \times \rho} \tag{1}$$

### 2.3.2. Electron Microscopy and Energy Dispersive X-ray Spectroscopy

Ultramicrotomy of the samples was performed before the samples were studied with (high-resolution) high-angle annular dark-field scanning transmission electron microscopy (HAADF-STEM). Samples were embedded in EpoFix resin and the embedded samples were left to harden in an oven at 60 °C overnight. Slices with a thickness of 70 nm were obtained by cutting the resin-embedded sample using a Reichert–Jung Ultracut E ultramicrotome with a Diatome Ultra 35° diamond knife. The sections were deposited on a glow-discharged carbon-formvar-coated copper grid (200 mesh). HAADF-STEM and Energy Dispersive X-ray Spectroscopy (EDX) were performed on an FEI Talos F200X electron microscope equipped with a Super-X EDX detector, operating at 200 kV. HR HAADF-STEM was performed on a Thermo Fisher Scientific Spectra 300 S/TEM at 300 kV. Scanning electron microscopy (SEM) on as-received zeotype samples was performed on a Helios G3 UC microscope operated at 10 kV.

### 2.3.3. X-ray Photoelectron Spectroscopy

X-ray photoelectron spectroscopy was performed using a KAlpha spectrometer from Thermo Fisher Scientific. The spectrometer was equipped with a monochromatic Al Kα X-ray source and a 180° double-focusing hemispherical analyzer. The samples were placed on double-sided carbon tape. Spectra were collected using an aluminium anode (Al, Kα = 1486.68 eV)

operating at 72 W and a spot size of 400 μm. Survey scans were measured at a constant pass energy of 200 eV. Region scans were measured at 50 eV. The shift in binding energy was calibrated to the C signal (285 eV). Each sample was measured four times. In order to obtain a quantitative value, Pd signals were normalized to the combined Si+Al areas, taking relative sensitivity factors into account.

### 2.3.4. Nitrogen Physisorption

Nitrogen physisorption isotherms were measured at −196 °C on a Micromeritics Tristar II Plus apparatus. First, the samples were dried overnight in a vacuum at 200 °C. The accessible surface areas were determined using the Brunauer–Emmett–Teller (BET), Dubinin–Astakhov and Langmuir methods. Micropore (<2 nm) volumes were determined using a Harkins–Jura thickness curve fitted between 0.32 and 0.40 nm thickness. The total pore volumes were derived from the amount of nitrogen adsorbed at $P/P_0 = 0.95$.

### 2.3.5. X-ray Diffraction

X-ray diffractograms were recorded using a Bruker-AXS D2 Phaser X-ray Diffractormeter in Bragg–Brentano mode, equipped with a Lynxeye detector (Co Kα1,2, λ = 1.790 Å). Recordings were taken at 2θ angles between 5° and 80° under constant rotation of 15 rpm.

### 2.4. Hydroconversion of n-Heptane and n-Hexadecane

Catalytic experiments were performed using an Avantium Flowrence 16-parallel fixed-bed reactor setup. Stainless steel reactors (inner diameter = 2 mm) were loaded with a bed height of 2 cm of SiC (220 μm), 30 mg of sieved catalyst (75–212 μm) and another layer of SiC, leaving 2 cm of free space at the top. The products were analyzed using an online GC (Agilent Technologies 7890B). The GC was equipped with an Agilent J&W HP-PONA column and the hydrocarbon products were analyzed using an FID. Prior to the catalytic tests with *n*-heptane (*n*-C7), catalysts were reduced at 350 °C in $H_2$/He (25 vol-% $H_2$). Experiments with *n*-heptane as feedstock were performed with the following reaction conditions: $H_2$/*n*-heptane molar ratio of 10, total pressure of 10 bar and WHSV of 2.4 $g_{n\text{-}C7} \cdot g_{cat}^{-1} \cdot h^{-1}$. Prior to catalytic tests with *n*-hexadecane (*n*-C16), catalysts were reduced at 350 °C in a pure hydrogen flow. Experiments with *n*-hexadecane as feedstock were performed with these reaction conditions: $H_2$/*n*-hexadecane molar ratio of 10, total pressure of 5 bar and WHSV of 2.9 $g_{n\text{-}C16} \cdot g_{cat}^{-1} \cdot h^{-1}$.

The conversion of *n*-heptane or *n*-hexadecane ($X_{n\text{-alkane}}$) is calculated by:

$$X_{n\text{-alkane}} = \left( 1 - \frac{F_{\text{Cwt.}n\text{-alkane,out}}}{F_{\text{Cwt.}n\text{-alkane,in}}} \right) \cdot 100\% \tag{2}$$

where in $F_{\text{Cwt.}n\text{-alkane,out}}$ and $F_{\text{Cwt.}n\text{-alkane,in}}$ are the flows based on carbon weight of *n*-alkane going out or in the reactor, respectively. The isomer yield ($Y_{i\text{-alkane}}$) is calculated by:

$$Y_{i\text{-alkane}} = \left( \frac{F_{\text{Cwt.}i\text{-alkane,out}}}{F_{\text{Cwt.}n\text{-alkane,in}}} \right) \cdot 100\% \tag{3}$$

Similarly, the yield of any cracked product is calculated by:

$$Y_{\text{Cm}} = \left( \frac{F_{\text{Cwt.Cm,out}}}{F_{\text{Cwt.}n\text{-alkane,in}}} \right) \cdot 100\% \tag{4}$$

in which Cm is a hydrocarbon molecule with m carbon atoms (m = 1–6 during *n*-heptane conversion and m = 1–14 during *n*-hexadecane conversion).

The selectivity towards *i*-heptane (*i*-C7) or cracked products (Cm) during *n*-heptane conversion is calculated using the following two equations:

$$S_{i\text{-}C7} = \left( \frac{F_{\text{Cwt.}i\text{-}C7,\text{out}}}{F_{\text{Cwt.}n\text{-alkane,in}} - F_{\text{Cwt.}n\text{-alkane,out}}} \right) \cdot 100\% \tag{5}$$

$$S_{\text{Cm}} = \left( \frac{F_{\text{Cwt.Cm, out}}}{F_{\text{Cwt.}n\text{-alkane,in}} - F_{\text{Cwt.}n\text{-alkane,out}}} \right) \cdot 100\% \tag{6}$$

The yield in mol-% of each product of *n*-hexadecane is calculated by:

$$Y_{\text{Cm}}(mol - \%) = \left( \frac{F_{\text{Cwt.Cm, out}}}{F_{\text{Cwt.}n\text{-alkane,in}}} \right) \cdot \left( \frac{16}{m} \right) \cdot 100\% \tag{7}$$

### 2.5. Calculation of Apparent Activation Energies

The apparent activation energies for the hydroconversion of *n*-heptane and *n*-hexadecane were calculated based on Arrhenius plots in which ln k was plotted against 1000/T. The results from the catalyst assessment were used to calculate ln k using Equation (8), assuming first order kinetics [44]:

$$\ln k = \ln\left( \frac{-\ln(1-X)}{\frac{W}{F}} \right) \tag{8}$$

In Equation (8), k is the first order rate constant (mol s$^{-1}$ kg$_{\text{cat}}$$^{-1}$), X is the conversion as a fraction, W is the weight of the catalyst (kg) and F is the molar flow of the reactant (mol s$^{-1}$). To stay in the kinetically determined regime, only ln k-values derived from X-values between 0.01 and 0.20 were plotted. At least five ln k-values were plotted against their corresponding 1000/T-values (1000 K$^{-1}$). The slopes of the Arrhenius plots were then multiplied by the gas constant R (8.314 J K$^{-1}$ mol$^{-1}$) to obtain the apparent activation energy (Ea, kJ mol$^{-1}$). Standard errors were derived from errors in the fit.

## 3. Results and Discussion

### 3.1. Acidity and Pd Weight Loading of Pd/SAPO-11/γ-Al$_2$O$_3$ and Pd/ZSM-22/γ-Al$_2$O$_3$ Catalysts

The Pd weight loading is 0.4–0.5 wt% on all catalysts (Table 1). This weight loading is sufficient to ensure that the reaction on the acid site is rate-determining, and slight differences in Pd loading do not impact the catalytic performance [45,46]. For a set of catalysts with the same zeotype, NH$_3$-TPD reveals a similar number of acid sites (Table 1). The pyridine-IR results also show the same number of Brønsted acid sites (BAS) within a set of catalysts with the same zeotype material (Table 1, Figure 1). For quantification of the number of BAS, the band at 1545 cm$^{-1}$ is assigned to the pyridinium ion. The band at 1453 cm$^{-1}$ is typical for coordinately bonded pyridine, indicating the presence of Lewis acid sites (LAS). The presence of the band at 1489 cm$^{-1}$ comes from pyridine adsorbed on both BAS and LAS and can therefore not be used for quantification of acid sites [41–43]. The band at 1576 cm$^{-1}$ can be assigned to pyridine on LAS and overlaps with a band of physisorbed pyridine [47–49]. Since the temperature of 250 °C should be sufficiently high for the desorption of pyridine, this band is most likely indicative of the presence of pyridine on LAS. Because of the rather identical acidity within a set of catalysts, the acidity is not expected to lead to differences in catalytic performance within a set of catalysts. An overview of the results obtained with NH$_3$-TPD and FT-IR of the catalysts, parent zeotypes and composite materials is presented in Table S1.

**Table 1.** Results of catalyst characterization: Pd weight loading, number of acid sites and Pd/(Si+Al) ratios.

| Catalyst | Pd Weight Loading [a] (wt%) | Number of Strong Acid Sites [b] (mmol/g) | Number of Brønsted Acid Sites (mmol/g) [c] | Pd/(Si+Al) Surface [d] (at/at) | Pd/(Si+Al) Bulk [a] (at/at) |
|---|---|---|---|---|---|
| Pd-γ-Al$_2$O$_3$/SAPO-11 | 0.49 | 0.29 | 0.015 | 0.0037 ± 0.0001 | 0.0036 |
| Pd-SAPO-11/γ-Al$_2$O$_3$ DR | 0.45 | 0.26 | n.d. | 0.0020 ± 0.0001 | 0.0032 |
| Pd-SAPO-11/γ-Al$_2$O$_3$ CR | 0.41 | 0.25 | 0.018 | 0.0018 ± 0.0001 | 0.0028 |
| Pd-γ-Al$_2$O$_3$/ZSM-22 | 0.50 | 0.37 | 0.023 | 0.0028 ± 0.0001 | 0.0026 |
| Pd-ZSM-22/γ-Al$_2$O$_3$ DR | 0.44 | 0.35 | n.d. | 0.0014 ± 0.0002 | 0.0023 |
| Pd-ZSM-22/γ-Al$_2$O$_3$ CR | 0.41 | 0.33 | 0.019 | 0.0010 ± 0.0001 | 0.0022 |

[a] Determined using inductively coupled plasma-optical emission spectroscopy on digested samples. [b] Determined using deconvolution of NH$_3$-TPD profiles and integration of the peak at higher temperatures (T ≥ 200 °C for SAPO-11 and T ≥ 300 °C for ZSM-22). [c] Determined using integration of the peak at 1545 cm$^{-1}$ of FT-IR spectra after adsorption and subsequent desorption of pyridine at 250 °C, correcting for the sample mass and using a molar absorption coefficient of 1.67 cm/μmol, (n.d. = not determined). [d] Determined from the areas of Pd, Si and Al peaks from XPS in four separate measurements, taking relative sensitivity factors into account.

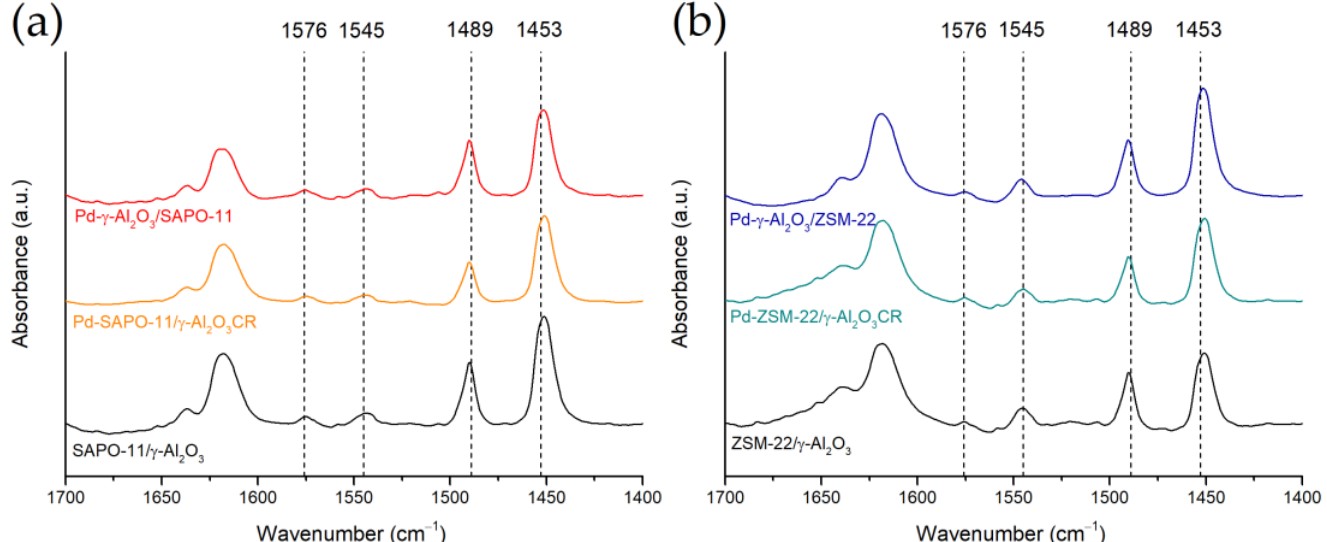

**Figure 1.** FT-IR spectra of the (**a**) SAPO-11 catalysts and (**b**) ZSM-22 catalysts after pyridine adsorption and subsequent desorption at 250 °C.

### 3.2. Determination of Pd Nanoparticle Location

The catalysts were embedded in a resin and 70 nm thick slices were prepared by ultramicrotoming. This enabled visualization of the interior of the catalyst particles. As can be seen in Figure 2, for the SAPO-11 catalysts, a rather clear distinction can be made between the Pd nanoparticles on the alumina binder (Figure 2a) and the Pd particles on/in the SAPO-11 crystals (Figure 2b,c). When Figure 2b,c are compared, more and larger Pd (~6–7 nm) nanoparticles are present on the outer surface of the SAPO-11 crystals when the catalysts undergo direct reduction (DR), whereas calcination-reduction (CR) results in more nanoparticles inside the SAPO-11 crystals.

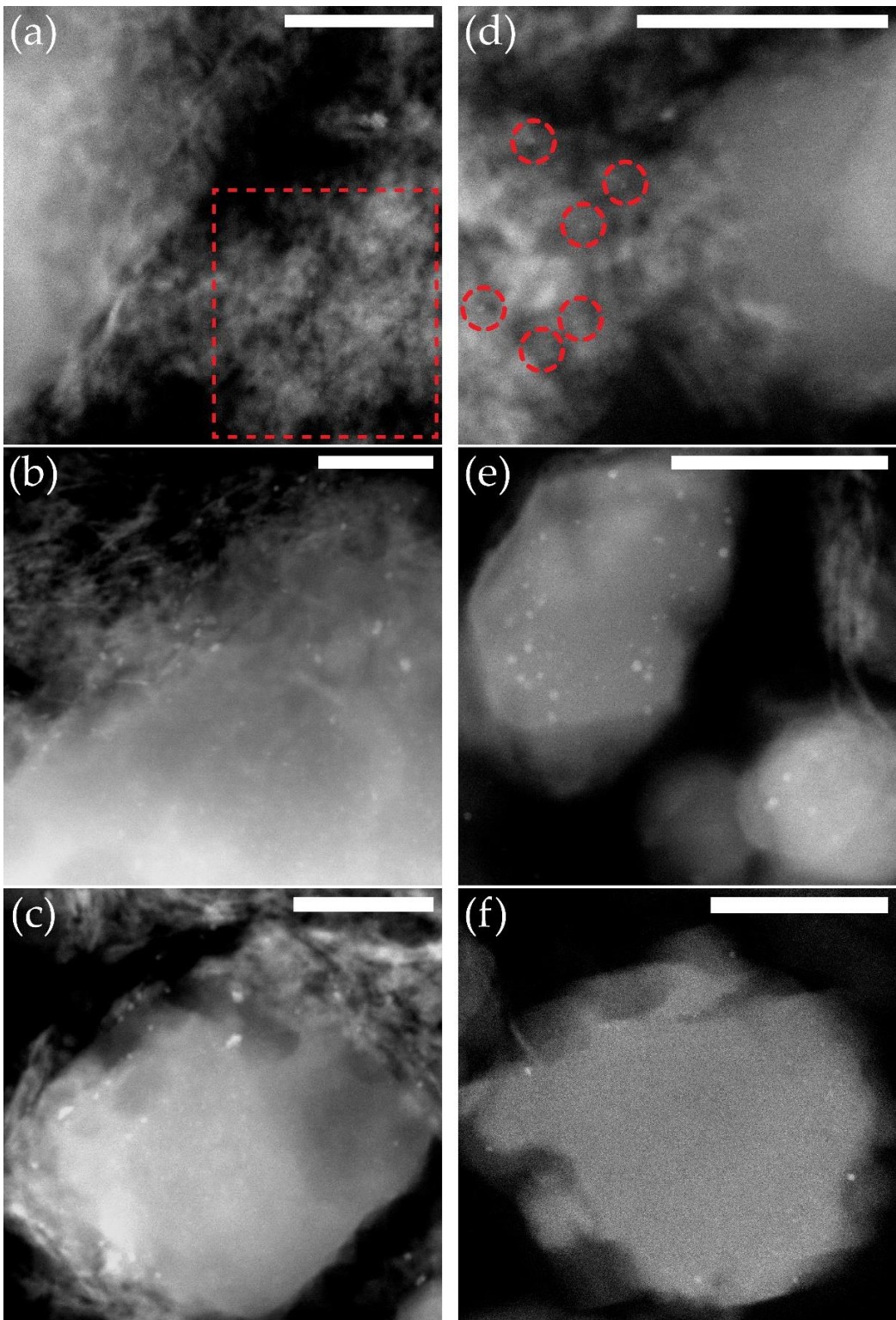

**Figure 2.** HAADF-STEM images of ultramicrotomed (thickness of 70 nm) catalyst samples: (**a**) Pd-γ-Al$_2$O$_3$/SAPO-11, (**b**) Pd-SAPO-11/γ-Al$_2$O$_3$ CR, (**c**) Pd-SAPO-11/γ-Al$_2$O$_3$ DR, (**d**) Pd-γ-Al$_2$O$_3$/ZSM-22, (**e**) Pd-ZSM-22/γ-Al$_2$O$_3$ CR, with cross sections of ZSM-22 crystals, (**f**) Pd-ZSM-22/γ-Al$_2$O$_3$ DR with cross sections of ZSM-22 crystals. Scalebars are 50 nm, red square and circles serve as guide to the eye towards Pd nanoparticles.

This also seems to be the case for the ZSM-22-based catalysts. After CR, more nanoparticles are inside the zeolite crystallite (Figure 2e) and after DR more nanoparticles are

located on the outer surface of the zeolite crystal and fewer particles are observed inside (Figure 2f). When it comes to the average nanoparticle size, only minor differences are observed (Table S2).

The imaging of Pd nanoparticles on the alumina binder (Figure 2a,d) is very challenging when the samples are ultramicrotomed, as the resolution and contrast are reduced by the presence of the resin. Therefore, it is assumed that the Pd nanoparticles did not change much upon mixing the Pd-$\gamma$-Al$_2$O$_3$ with the respective zeotype materials. An HAADF-STEM image of the original Pd-$\gamma$-Al$_2$O$_3$ is included in Figure S2.

X-ray photoelectron spectroscopy (XPS) studies were performed to further elaborate on the metal nanoparticle location. Since XPS is a surface-sensitive technique, only Pd atoms residing on the surface of the zeotype crystals or on the alumina binder contribute to the signal in the XPS. When Pd atoms are deeper inside the zeotype crystals, the emitted electrons will not escape from the sample, resulting in a low XPS signal.

In both catalyst series, the highest Pd signal is obtained when Pd is on the alumina binder (Table 1, Supplementary Figure S3). When Pd is on the zeotype, a lower Pd/(Si+Al) signal is observed. This can have two causes: (1) typically the Pd nanoparticles after direct reduction are larger [34], which results in relatively fewer Pd atoms contributing to the Pd signal, because only the outer layers of the nanoparticle contribute, and (2) only the Pd nanoparticles that are on the outside contribute to the signal, whereas the particles that reside inside the zeotype crystals will not contribute to the signal. The Pd/(Si+Al) is lowest in both the SAPO-11- and ZSM-22-based catalysts after calcination and reduction. The dispersion in the SAPO-11-based catalysts is higher after calcination-reduction than after DR (Table S2), so more Pd atoms can contribute to the XPS signal if they are on the outer surface of the SAPO-11 crystals. For the catalysts with Pd in/on ZSM-22, the dispersions are very similar. Combining these results shows that the calcination-reduction method results in the largest amount of Pd residing inside the zeotype crystals.

Moreover, based on the N$_2$ physisorption measurements of the samples containing no alumina, the CR procedure indeed leads to more pore blocking than DR, decreasing the BET surface area (Table S3). This is the case even while the DR samples have higher amounts of Pd than the CR samples used for these measurements.

Although CR and DR clearly affect the Pd nanoparticle location as observed with HAADF-STEM and XPS, STEM-EDX mapping reveals Pd nanoparticles on the outer surface of SAPO-11 in the Pd-SAPO-11/$\gamma$-Al$_2$O$_3$ CR catalysts as well (Figure 3). In these catalysts, some Pd nanoparticles are present on silica, which is not in the SAPO-11 framework. The placement of Pd on this extraframework silica diminishes the differences in Pd nanoparticle location between the SAPO-11-based catalysts.

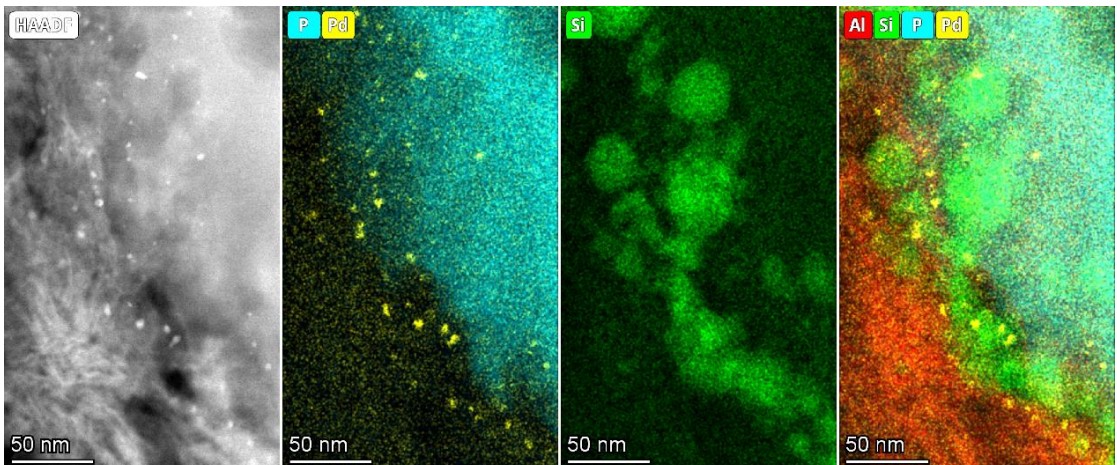

**Figure 3.** HAADF-STEM image and EDX-maps of an ultramicrotomed sample of reduced Pd-SAPO-11/$\gamma$-Al$_2$O$_3$ catalyst prepared with the calcination-reduction (CR) method. The elements Pd, P, Si and Al are depicted in yellow, blue, green and red, respectively.

### 3.3. The Effect of the Metal Nanoparticle Location on n-Heptane Conversion

The effect of the different catalyst preparation methods and hence the location was assessed during the hydroconversion of *n*-heptane. As can be seen in Figure 4a, the preparation method has only a limited impact on the catalytic activity of the different SAPO-11-based catalysts. Considering the selectivity (Figure 4b), at conversions above 60% the catalyst with Pd inside the SAPO-11 crystals (Pd-SAPO-11/γ-Al$_2$O$_3$ CR) is somewhat less selective towards *i*-heptane. This is in line with findings from previous works where lower isomer selectivity (and hence more cracking) is observed when Pt is in the zeolite crystals compared to catalysts with Pt on the alumina binder [16,20,21], provided that the metal weight loading is high enough [22]. Surprisingly, the activity is slightly lower and the isomer selectivity is higher when the Pd nanoparticles are on the SAPO-11 crystals, which is the case for the Pd-SAPO-11/γ-Al$_2$O$_3$ DR catalyst.

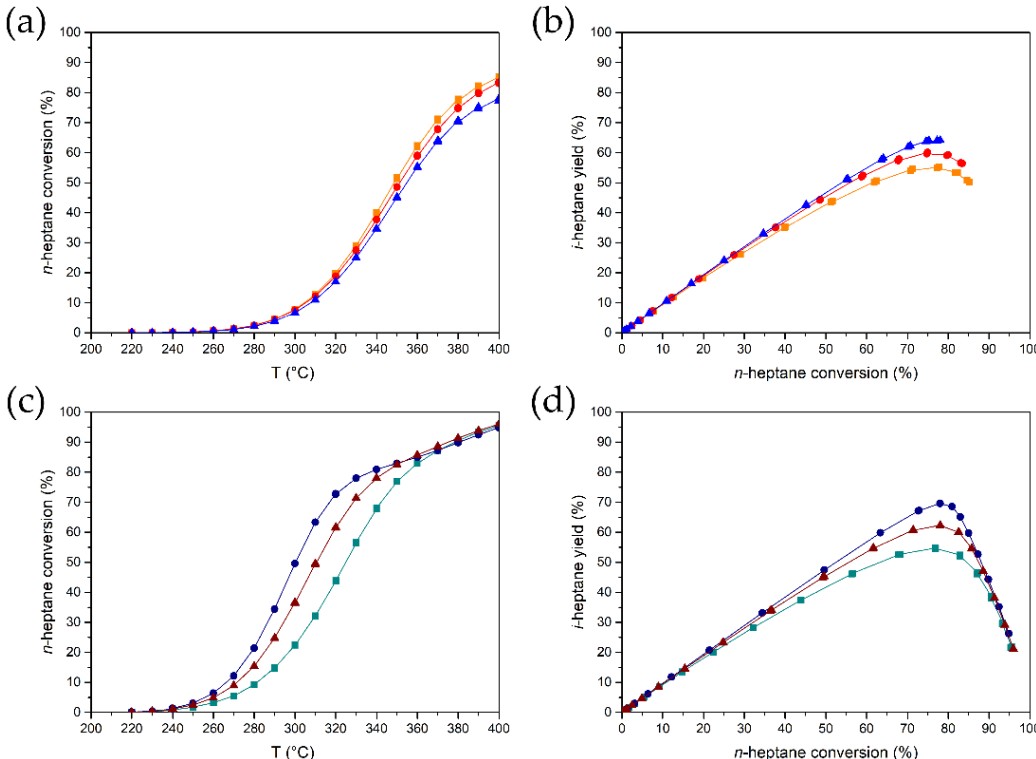

**Figure 4.** Results of the hydroconversion of *n*-heptane. (**a**) *n*-heptane conversion as function of the temperature and (**b**) *i*-heptane yield as function of the *n*-heptane conversion for Pd-SAPO-11/γ-Al$_2$O$_3$ CR (orange squares), Pd-γ-Al$_2$O$_3$/SAPO-11 (red circles) and Pd-SAPO-11/γ-Al$_2$O$_3$ DR (blue triangles). (**c**) *n*-heptane conversion as function of the temperature and (**d**) *i*-heptane yield as function of *n*-heptane conversion for Pd-ZSM-22/γ-Al$_2$O$_3$ CR (cyan squares), Pd-γ-Al$_2$O$_3$/ZSM-22 (dark blue circles) and Pd-ZSM-22/γ-Al$_2$O$_3$ DR (brown triangles).

When the hydroconversion of the ZSM-22-based catalysts is considered (Figure 4c,d), there are more distinct differences in the catalytic performance caused by a difference in the Pd nanoparticle location. Placing Pd on the alumina binder is clearly beneficial for the catalytic activity and selectivity. The catalytic activity and isomer selectivity is the lowest when Pd is located inside the ZSM-22 crystals, as is the case for Pd-ZSM-22/γ-Al$_2$O$_3$ CR catalysts. Interestingly, placing Pd particles on the outer surface of the ZSM-22 crystals by means of direct reduction can already partially induce the benefits of placing Pd on the alumina binder, as is shown by the in-between activity and selectivity of the Pd-ZSM-22/γ-Al$_2$O$_3$ DR catalyst. This result is further evidence for the effect of the heat treatment on the Pd location.

However, unlike the results obtained with the SAPO-11-based catalysts, the DR catalyst shows intermediate performance in the ZSM-22 case. A tentative explanation is that

after DR, there are still Pd nanoparticles present in the zeotype micropores, impeding intracrystalline diffusion and enlarging residence times, which causes (over)cracking. The differences in the maximum isomer yield in the SAPO-11-based catalysts are small compared to the differences exhibited by the ZSM-22 catalysts. This may be caused by the presence of Pd nanoparticles on extraframework silica, limiting the difference in Pd nanoparticle location in the SAPO-11-based catalysts.

### 3.4. The Effect of the Metal Nanoparticle Location on n-Hexadecane Conversion

If we compare the conversion of *n*-hexadecane for all SAPO-11-based catalysts (Figure 5a), the activity of the three catalysts is rather similar although at the highest temperatures some differences are apparent. Furthermore, if *i*-hexadecane selectivity is considered (Figure 5b), the highest isomer yield is observed when Pd is on the alumina binder. This is followed by the DR catalyst and the lowest isomer yield is obtained when the catalyst with Pd inside the SAPO-11 crystals is used. This is again in line with the results from previous work [16,20,21,26]. However, it slightly differs from the *n*-heptane results in which the DR catalyst shows the highest *i*-heptane selectivity. This difference can be attributed to the fact that fewer Pd nanoparticles are outside the SAPO-11 crystals in the DR catalyst compared to the catalyst with Pd on the alumina binder. The diffusivity of the longer molecule towards the Pd nanoparticles inside the crystals is impeded; hence the ratio between the available Pd sites and acid sites is lower than the overall $n_{Pd}/n_a$ ratio. This is in line with previous results, which suggest that a higher $n_m/n_a$ ratio is required to obtain the desired products during the conversion of longer hydrocarbon feedstocks [37]. Note that SAPO-11-based catalysts deactivate at higher temperatures. This may be caused by the presence of impurities in the SAPO-11 samples as evidenced by differences in the XRD patterns (Figure S4), SEM (Figure S5) and the aforementioned presence of extraframework silica (Figure 3).

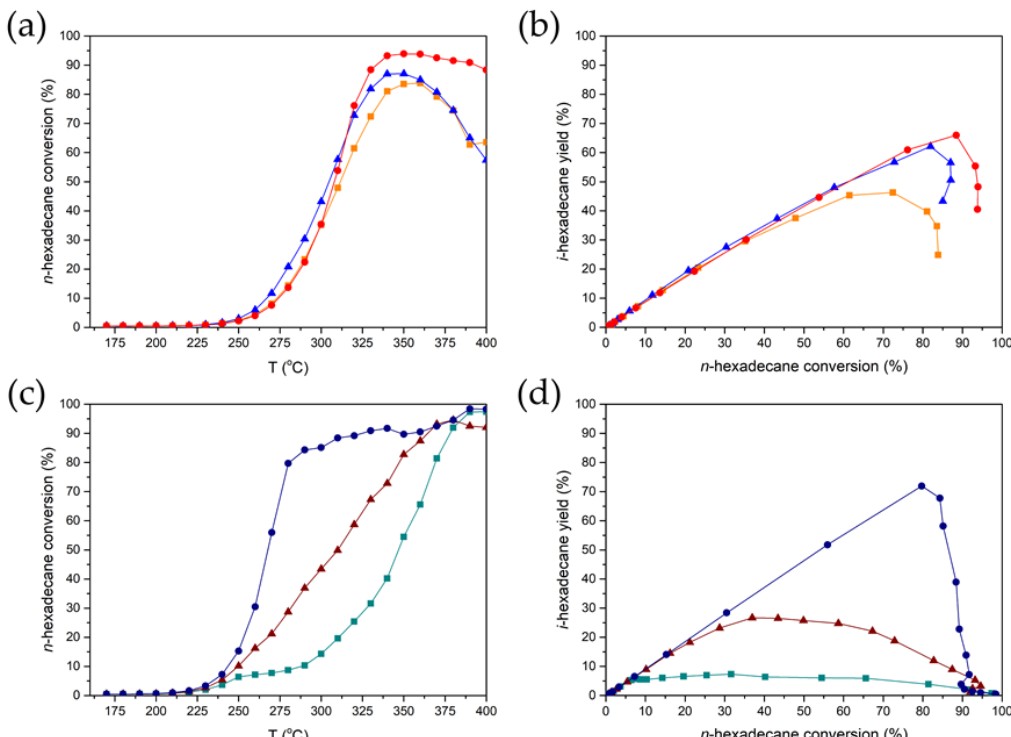

**Figure 5.** Results of the hydroconversion of *n*-hexadecane. (**a**) *n*-hexadecane conversion as function of the temperature and (**b**) *i*-hexadecane yield as function of the *n*-hexadecane conversion for Pd-SAPO-11/γ-Al$_2$O$_3$ CR (orange squares), Pd-γ-Al$_2$O$_3$/SAPO-11 (red circles) and Pd-SAPO-11/γ-Al$_2$O$_3$ DR (blue triangles). (**c**) *n*-hexadecane conversion as function of the temperature and (**d**) *i*-hexadecane yield as function of *n*-hexadecane conversion for Pd-ZSM-22/γ-Al$_2$O$_3$ CR (cyan squares), Pd-γ-Al$_2$O$_3$/ZSM-22 (dark blue circles) and Pd-ZSM-22/γ-Al$_2$O$_3$ DR (brown triangles).

For the ZSM-22-based catalysts, the catalyst with Pd on the alumina binder (Pd-$\gamma$-Al$_2$O$_3$/ZSM-22) displays the highest activity and $i$-hexadecane selectivity (Figure 5c,d). This catalyst is by far the most selective with a maximum isomer yield of over 70%. When Pd is mainly present inside the ZSM-22 crystals (Pd-ZSM-22/$\gamma$-Al$_2$O$_3$ CR), the isomer yield does not even reach 10%. This suggests that the majority of the Pd particles in this catalyst are inside the ZSM-22 crystals, thus giving rise to overcracking. The elevated isomer yield of the Pd-ZSM-22/$\gamma$-Al$_2$O$_3$ DR catalyst of about 30% shows that the DR results in a larger fraction of Pd on the outside of the ZSM-22 catalysts, allowing for more isomerization to occur. Note that at low conversion levels (< 5%), the isomer yields are very similar, whereas at higher conversion levels the isomer yields differ substantially.

### 3.5. The Effect of the Metal Nanoparticle Location on Hydrocracking Selectivities

The cracking patterns are compared by plotting the selectivity for each product against the product carbon number. Figure 6 shows the product selectivities over the Pd/zeotype/$\gamma$-Al$_2$O$_3$ catalysts during both $n$-heptane and $n$-hexadecane conversion. Next to isomerization, the main reactions are acid cracking, hydrogenolysis and dimerization cracking [21]. During the hydroconversion of $n$-heptane, the main products that are formed next to $i$-heptane are propane, $n$-butane and $i$-butane (Figure 6a,b). As $i$-butane and $n$-butane are products of different $\beta$-scission mechanisms [6], their respective ratios tell us more about the mechanisms involved. For SAPO-11-based catalysts, $n$-butane yields are very low, resulting in high $i$-butane/$n$-butane ($i$-C4/$n$-C4) ratios. It shows that in all three catalysts, hydrocracking mainly occurs via the type B $\beta$-scission mechanism. This is a result of the relatively mild acidity of SAPO-11 [50,51], decreasing its ability to catalyze the energetically less favorable type C $\beta$-scission. ZSM-22 catalysts, on the contrary, produce less $i$-butane, with $i$-C4/$n$-C4 ratios ranging between 1.3 and 1.7, approaching a 50:50 ratio. This indicates that type B and type C $\beta$-scission mechanisms [6,52] occur to a rather similar extent. The higher acidity of ZSM-22 allows for more type C $\beta$-scission producing $n$-butane. Furthermore, the $i$-C4/$n$-C4 ratios for this set of catalysts increases when more Pd nanoparticles are assumed to be outside the ZSM-22 crystals, which is in line with previously obtained results [16,21]. When more metal nanoparticles are inside the zeolite crystals, the cracking reactions are more prone to confinement effects, also known as shape-selectivity, resulting in improved $n$-butane yields.

For the hydrocracking of $n$-hexadecane, as shown in Figure 6c,d, the more symmetrical these plots are, i.e., the more 'bell-shaped' the curve is (for every C10 molecule, a C6 molecule is formed and for every C8 molecule another C8 molecule is formed, etc.), the more "ideal" hydrocracking takes place [6]. Ideal hydrocracking means that a catalyst tends to perform only one cracking step. If the selectivity peak is shifted towards lower carbon numbers, the catalyst performs more secondary cracking, resulting in the production of more shorter hydrocarbons. In practice, this is undesired as overcracking of the feedstock yields small hydrocarbons not suitable for gasoline, jet fuel or diesel applications. Although SAPO-11-based catalysts show relatively high isomer selectivity, the cracked products mainly originate from secondary cracking as can be observed in Figure 4c. This may be caused by the shape and size of the SAPO-11 pores (10MR, 0.39 nm $\times$ 0.63 nm). When hexadecane isomers reside inside these pores, diffusivities are very low, resulting in high residence time and the increased probability of secondary reactions such as cracking.

For both Pd-ZSM-22 catalysts, clearly not all Pd is outside the zeolite crystals as the selectivity towards the cracked products is still high and the isomer yield of only 30% is much lower than the isomer yield obtained when the Pd nanoparticles are on the alumina binder. As can be seen in Figure 6d, the Pd nanoparticle location impacts the cracking behavior of the ZSM-22-based catalysts. The catalyst with Pd on the alumina binder acts more like an "ideal" hydrocracking catalyst, as the plot is most similar to a bell-shaped carbon distribution [6]. If more of the Pd particles are present inside the ZSM-22 crystals—as is the case to some extent for the DR catalyst and even more for the Pd-ZSM-22/$\gamma$-Al$_2$O$_3$ CR catalyst—more secondary cracking occurs.

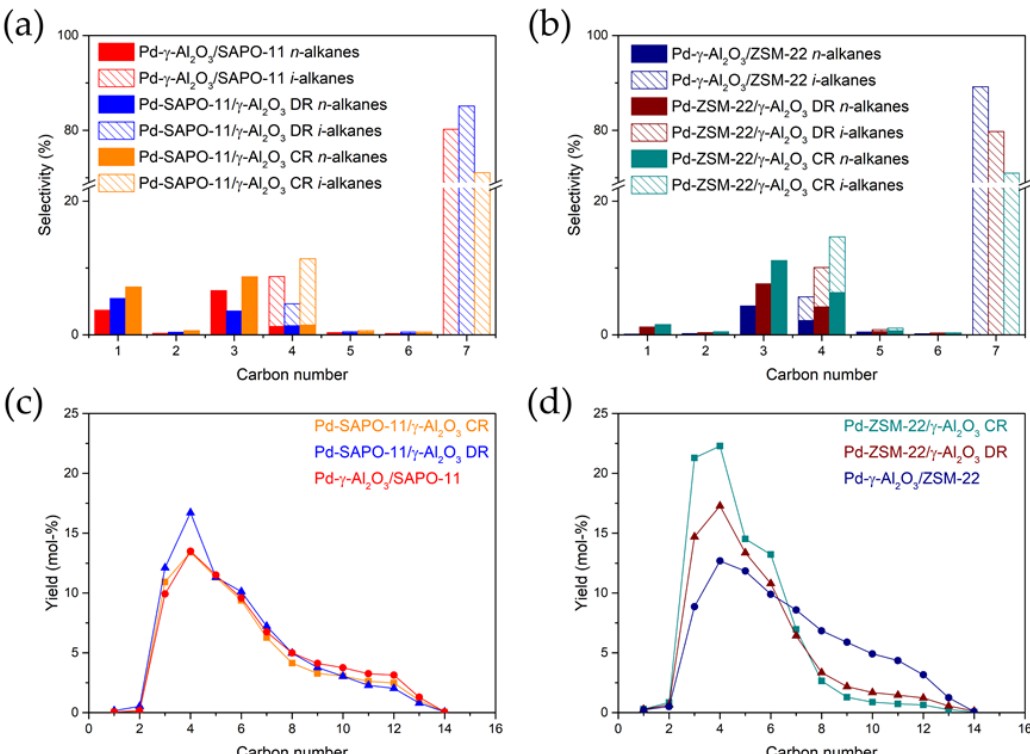

**Figure 6.** Distribution of products obtained during (**a**) hydroconversion of *n*-heptane over SAPO-11-based catalysts at 75–77% conversion, (**b**) hydroconversion of *n*-heptane over ZSM-22-based catalysts at 77–78% conversion, (**c**) hydroconversion of *n*-hexadecane over Pd-SAPO-11/γ-Al$_2$O$_3$ (orange squares at T = 340 °C), Pd-γ-Al$_2$O$_3$/SAPO-11 (red circles at T = 350 °C) and Pd-SAPO-11/γ-Al$_2$O$_3$ DR (blue triangles at T = 380 °C) at 26–28% cracking yield and (**d**) hydroconversion of *n*-hexadecane over Pd-ZSM-22/γ-Al$_2$O$_3$ (cyan squares at T = 340 °C), Pd-γ-Al$_2$O$_3$/ZSM-22 (dark blue circles at T = 320 °C) and Pd-ZSM-22/γ-Al$_2$O$_3$ DR (brown triangles at T = 330 °C) at 25–32% cracking yield.

In Table 2, an overview of the apparent activation energies for hydroconversion over the different catalysts is presented. These activation energies are derived from Arrhenius plots (Figure S6). The activation energies in the set of SAPO-11-based catalysts are virtually identical, indicating that there is no difference in diffusivity as a result of a difference in Pd location. This is in line with the fact that the catalytic activity of these catalysts is not impacted by the Pd location, possibly caused by the presence of Pd particles on the extraframework silica in the SAPO-11 catalysts. Furthermore, this extraframework silica can have a rate-enhancing effect, increasing the transport of molecules of which the intracrystalline diffusion is not limited [53]. This possibly results in similar diffusion rates for all three catalysts.

For the ZSM-22-based catalysts, however, the Pd location has a great impact on the activity and the apparent activation energy of both *n*-heptane and *n*-hexadecane conversion. This difference in activation energy is most probably a result of the diffusion effects caused by a difference in the Pd nanoparticle location [21]. Here, the DR catalyst shows activation energies that are in between the activation energies of the Pd-on-alumina and the Pd-in-ZSM-22 catalysts, indicating that as a result of the DR treatment indeed more of the Pd nanoparticles are located *on* the ZSM-22 crystals.

**Table 2.** Apparent activation energies ($E_a$) for the hydroconversion of *n*-heptane and *n*-hexadecane over each catalyst.

| Catalyst | $E_a$ *n*-Heptane Conversion (kJ mol$^{-1}$) | $E_a$ *n*-Hexadecane Conversión (kJ mol$^{-1}$) |
|---|---|---|
| Pd-$\gamma$-Al$_2$O$_3$/SAPO-11 | $148 \pm 1$ | $143 \pm 4$ |
| Pd-SAPO-11/$\gamma$-Al$_2$O$_3$ DR | $152 \pm 1$ | $140 \pm 2$ |
| Pd-SAPO-11/$\gamma$-Al$_2$O$_3$ CR | $150 \pm 1$ | $140 \pm 1$ |
| Pd-$\gamma$-Al$_2$O$_3$/ZSM-22 | $173 \pm 4$ | $148 \pm 8$ |
| Pd-ZSM-22/$\gamma$-Al$_2$O$_3$ DR | $160 \pm 4$ | $119 \pm 2$ |
| Pd-ZSM-22/$\gamma$-Al$_2$O$_3$ CR | $135 \pm 1$ | $85 \pm 1$ |

Overall, the effect of the heat treatment during the preparation of the Pd-ZSM-22 catalysts is more profound than in the Pd-SAPO-11 catalysts. This effect is found in all product distributions. The apparent activation energies for both *n*-heptane and *n*-hexadecane conversion are also largely impacted by the Pd nanoparticle location in the ZSM-22-based catalysts, whereas they are not affected in the SAPO-11-based catalysts (Table 2). This can be a result of the fact that in general, more of the Pd particles end up on the outside of the SAPO-11 crystals regardless of the heat treatment as evidenced by the comparison of the Pd/(Si+Al) surface versus bulk ratios (Table 1) and the presence of Pd on extraframework silica revealed with EDX (Figure 3).

**4. Conclusions**

The heat treatment of zeotypes that underwent IE with Pd(NH$_3$)$_4$(NO$_3$)$_2$ impacted the final location of the Pd nanoparticles with respect to the acid sites. Although in both cases, metal nanoparticles seemed to be inside and on the zeotype crystals, our results showed that a slow calcination followed by a reduction resulted in more Pd particles inside the zeotype crystals. Direct reduction, on the other hand, caused more Pd nanoparticles to end up on the zeotype crystals. This was further investigated using TEM and XPS. For the SAPO-11-based catalysts, HAADF-STEM showed to be a tool to investigate the differences in location. The XPS revealed more clear differences in nanoparticle location for the ZSM-22 catalysts. In turn, the hydroconversion of *n*-heptane and *n*-hexadecane clearly showed differences in Pd nanoparticle location, as having more Pd nanoparticles on the outside was beneficial for the activity and selectivity towards the isomers for both sets of catalysts. Having Pd on the alumina binder was still the most beneficial for the hydroconversion performance. Direct reduction could be used to prepare catalysts with more Pd on the outer surface of the zeotype crystals, resulting in an intermediate hydroconversion performance. The calcination-reduction treatment was a strategy to encapsulate Pd nanoparticles inside the crystals of the zeolitic materials, and was disadvantageous for hydroconversion purposes.

**Supplementary Materials:** The following supporting information can be downloaded at: https://www.mdpi.com/article/10.3390/chemistry5010026/s1, Figure S1: Schematic overview of catalyst synthesis; Table S1: Number of acid sites as determined using NH$_3$-TPD and FT-IR using pyridine (Py) as probe molecule; Table S2: Palladium loading, palladium nanoparticle properties and acid properties of each catalyst; Figure S2: HAADF-STEM image of Pd/$\gamma$-Al$_2$O$_3$ before mixing with a zeotype material; Figure S3: XPS spectra of Pd in all catalysts; Table S3: Results of N$_2$ physisorption on samples without $\gamma$-alumina; Figure S4: XRD patterns; Figure S5: SEM images; Figure S6: Arrhenius plots.

**Author Contributions:** Conceptualization, L.C.J.S., S.T.R., G.J.S. and K.P.d.J.; Methodology, L.C.J.S., J.H.v.d.M. and K.C.; Validation, L.C.J.S. and J.H.v.d.M.; Formal Analysis, L.C.J.S., J.H.v.d.M. and A.L.; Investigation, L.C.J.S., J.H.v.d.M., J.D.M., M.T. and A.L.; Resources, K.P.d.J., P.E.d.J. and E.J.M.H.; Data Curation, L.C.J.S.; Writing—Original Draft Preparation, L.C.J.S.; Writing—Review and Editing, J.H.v.d.M., A.L., S.T.R., G.J.S., E.J.M.H., P.E.d.J. and K.P.d.J.; Supervision, K.P.d.J. and P.E.d.J.; Project Administration, L.C.J.S.; Funding Acquisition, P.E.d.J. and K.P.d.J. All authors have read and agreed to the published version of the manuscript.

**Funding:** This research was funded by BP plc.

**Data Availability Statement:** Original data are available upon request.

**Acknowledgments:** The authors thank Dennie Wezendonk, Jan Willem de Rijk and Remco Dalebout for their help commissioning the Flowrence setup. Claudia Keijzer is thanked for SEM imaging. Remco Dalebout, Laura Barberis, Kristiaan Helfferich and Suzan Schoemaker are acknowledged for performing the N$_2$ physisorption measurements.

**Conflicts of Interest:** The authors declare no conflict of interest.

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
