# Peer review of "Steering the Metal Precursor Location in Pd/Zeotype Catalysts and Its Implications for Catalysis"

_chemistry, doi:10.3390/chemistry5010026_

Round 1

Reviewer 1 Report

The present form of the manuscript is difficult to follow as many figures and tables are given as Supplementary material. If it is not possible to revise manuscript by adding more figures to the main text, explain the figure in words to make it more readable. 

Line 317: "... is rather similar as can be seen ..." similar to what?

Table S1: FTIR results for direct reduction (DR) treatment catalysts were not shown. Explain. 

Fig. S1: If only two FTIR peaks are followed to show BAS and LAS, what is the third peak located in-between the two, and why is it labelled? 

Fig. S3: Very noisy XPS spectra, signal/noise ratio is too low to resolve Pd 3d bands. Discussion and Tables involving the peaks should be accompanied with uncertainty levels.  

Reviewer 2 Report

The submitted paper presents the interesting results concerning bifunctional catalysts and the effect of the heat treatment of the catalyst on Pd nanoparticles location and finally on the activity of catalysts in n-heptane and n-hexadecane  hydroconversion.

The most of experiments were done with care and the interpretations are sound with respect to the experimental evidences. The paper is well written and there is no doubt that the results are worthy for publication.

However, I have one remark concerning IR experiment with Pyridine adsorption. Authors calculated the number of LAS from the spectra obtained after evacuation of Py at 150C (Table S1). In my opinion the spectra taken after Py desorption at higher temperature (200 or 250 C) should be taken into account. The bands at ca. 1570 cm-1 are clearly visible in the spectra of catalysts (Figure S1). This band, together with the band ca. 1440 cm-1, is characteristic of physisorbed pyridine. It indicates that the band from LAS is disturbed by the band from Py physisorbed on the surface of catalyst. 

Reviewer 3 Report

The manuscript deals with the preparation of different type of Pd catalysts based on ZSM-22 and SAPO-11 zeolites. Overall, I think the authors made a solid study and achieved their goal, with some minor modifications the manuscript can be published.

1. The experimental part (especially 2.2 catalyst synthesis) is overloaded. I suggest to split it in a few parts and add visual synthetic procedures because in the current state it is not easy to perceive.

2. Why the authors chose these two materials? They are quite different. Was that a random choice?

3. The concertation of strong acid sites was determined by NH3-TPD. For ZSM-22 and SAPO-11 the deconvolution was different - T ≥ 200 °C for SAPO-11 and T ≥ 300 °C for ZSM-22. I understand why the authors did that way, however, it was not entirely correct. I suggest to use IR of adsorbed pyridine to obtain relevant data.
